# *Conyza sumatrensis* Resistant to Paraquat, Glyphosate and Chlorimuron: Confirmation and Monitoring the First Case of Multiple Resistance in Paraguay

**Alfredo Junior Paiola Albrecht [1], Guilherme Thomazini [2], Leandro Paiola Albrecht [1], Afonso Pires [3], Juliano Bortoluzzi Lorenzetti [4], Maikon Tiago Yamada Danilussi [4], André Felipe Moreira Silva [5],* and Fernando Storniolo Adegas [6]**

[1] Department of Agronomic Sciences, Federal University of Paraná, Palotina, PR 85950-000, Brazil; ajpalbrecht@yahoo.com.br (A.J.P.A.); lpalbrecht@yahoo.com.br (L.P.A.)

[2] Department of Agronomic Sciences, Maringá State University, Umuarama, PR 87502-970, Brazil; guilherme_thomazini@hotmail.com

[3] Semillas Pires, Corpus Christi, Canindeyú 7850, Paraguay; afonsopirespy@gmail.com

[4] Department of Crop Science and Phytosanitary, Federal University of Paraná, Curitiba, PR 80060-000, Brazil; lorenzettijb@gmail.com (J.B.L.); maikondanilussi@gmail.com (M.T.Y.D.)

[5] Crop Science, Palotina, PR 85950-000, Brazil

[6] Embrapa Soybean, Londrina, PR 86001-970, Brazil; fernando.adegas@embrapa.br

\* Correspondence: afmoreirasilva@alumni.usp.br

**Abstract:** *Conyza sumatrensis* was reported to be associated with 20 cases of herbicide resistance worldwide, with a recent report of multiple drug resistance to paraquat, glyphosate, and chlorimuron in Brazil. In Paraguay, there were no reports of cases of resistance for this species; however, in 2017, researchers began identifying biotypes with resistance to paraquat, glyphosate, and chlorimuron, which is the focus of the present study. The goal of this study was to investigate the case of multiple resistance of *C. sumatrensis* to paraquat, glyphosate, and chlorimuron and to monitor the resistant biotypes in the departments of Canindeyú and Alto Paraná. Seeds were collected from sites where plants survived after herbicide application in the 2017/18 and 2018/19 seasons. After screening, biotypes were selected for the construction of dose–response curves. A resistance factor (RF) of 6.79 was observed for 50% control ($C_{50}$) and 3.92 for 50% growth reduction ($GR_{50}$) for the application of paraquat. An RF of 12.32 was found for $C_{50}$ and 4.15 for $GR_{50}$ for the application of glyphosate. For the application of chlorimuron, an RF of 11.32 was found for $C_{50}$ and 10.96 for $GR_{50}$. This confirms the multiple resistance of the *C. sumatrensis* biotype to paraquat, glyphosate, and chlorimuron. Population monitoring indicated the presence of *C. sumatrensis* with multiple resistance in departments of Canindeyú and Alto Paraná, Paraguay.

**Keywords:** ALS inhibitors; EPSPs inhibitors; herbicides; herbicide-resistance; South America; Sumatran fleabane; photosystem I inhibitor; weeds

## 1. Introduction

The selection of herbicide-resistant weed biotypes is one of the major problems in agriculture today. Herbicide resistant biotypes of hairy fleabane (*Conyza bonariensis* L. Cronquist), horseweed (*Conyza canadensis* L. Cronquist) and Sumatran fleabane (*Conyza sumatrensis* (Retz.) E. Walker) were previously reported. There are currently 105 herbicide-resistance cases for the three *Conyza* spp., which includes resistance to 5-enolpyruvylshikimate-3-phosphate synthase (EPSPs) inhibitors, acetolactate synthase (ALS) inhibitors, synthetic auxins, and photosystem I inhibitors, among others [1].

The weed *Conyza* spp. is among the most problematic in soybean crops in Paraguay. Regarding the impact of this species on crops, Trezzi et al. [2] indicated that 2.7 plants of *Conyza* spp. $M^{-2}$ can reduce the soybean yield by 50%. The species has an annual life cycle, herbaceous size, and high seed production and is found in several agricultural environments, such as grain crops [3,4]. *Conyza sumatrensis* is believed to be originally from the subtropical region of South America, with dispersion to Europe, America, and Asia [5,6]. A single plant can produce more than 200 thousand seeds, which germinate mainly from fall to early spring [7].

Among the factors that lead to the selection of herbicide-resistant weed biotypes, there is the use of the same herbicides, or different herbicides but with the same mode of action, in which strong selection pressure results in the selection of resistant biotypes [8–10]. For soybean crops in Paraguay, one of the most common management techniques for *Conyza* spp. is the application, in the off-season, of glyphosate + 2,4-D with paraquat in sequence, in some cases with the application of diclosulam at soybean pre-emergence. In post-emergence, the application of glyphosate alone or in mixtures with ALS-inhibiting herbicides may be used.

One of the main tools for delaying the selection of resistant weed biotypes, as well as managing plants with cases of resistance, is the diversification of management practices, with an emphasis on the rotation and combination of herbicides integrated with non-chemical measures. In this context, monitoring resistant weed populations allows for the identification of the evolution and dispersion of resistance cases, which consequently provides important information for decision making for weed control [11–13].

*Conyza sumatrensis* presents 20 cases of herbicide resistance worldwide, and seven in Brazil [1], with a recent report of multiple resistance to glyphosate, chlorimuron, and paraquat [14]. In Paraguay, there were no reports of cases of resistance for this species; however, in 2017, researchers began focusing on identifying biotypes with resistance to glyphosate (an EPSPs inhibitor), chlorimuron (an ALS inhibitor), and paraquat (a photosystem I inhibitor) [1]. This was reported to the International Herbicide-Resistant Weed Database, and registered in it, and is the focus of the present study. Thus, the aim of this study was to investigate the case of triple resistance of *C. sumatrensis* to the herbicides glyphosate, chlorimuron, and paraquat, and to monitor resistant biotypes mainly in the departments of Canindeyú and Alto Paraná, in Paraguay.

## 2. Material and Methods

### 2.1. Seed Collection

Seeds were collected in sites where *C. sumatrensis* plants survived after herbicide burndown application in pre-sowing in the 2017/18 and 2018/19 growing seasons, in 33 agricultural areas located in the departments of Canindeyú and Alto Paraná, Paraguay. The geographical coordinates, biotype identification, and infested crops are listed in Table 1. Among these sites, there are two locations with possibly susceptible plants that served as a comparison control.

The sampling sites were chosen according to reports of control failures as sites with possible cases of resistance. Our seed collection followed the methodology proposed by Burgos et al. [15]. For each site, seeds were collected from 5–10 plants, with the same characteristics, pooled into a single sample per site (with at least 1000 physiologically mature seeds per sample).

**Table 1.** Collection sites of *Conyza sumatrensis* populations with suspected resistance to herbicides. Canindeyú and Alto Paraná, Paraguay, 2017/18 and 2018/19 seasons.

| Site | Department | Latitude | Longitude | Crop |
|------|-----------|----------|-----------|------|
| 01 | Canindeyú | 24°10′30″ S | 54°41′33″ W | Soybean |
| 02 | Canindeyú | 24°15′28″ S | 54°47′30″ W | Soybean |
| 03 | Canindeyú | 24°14′42″ S | 54°52′25″ W | Soybean |
| 04 | Canindeyú | 24°13′18″ S | 54°52′09″ W | Soybean |
| 05 | Canindeyú | 24°20′09″ S | 54°49′41″ W | Soybean |
| 06 | Canindeyú | 24°22′23″ S | 54°50′26″ W | Soybean |
| 07 | Canindeyú | 24°23′59″ S | 54°50′54″ W | Soybean |
| 08 | Canindeyú | 24°34′53″ S | 54°51′39″ W | Soybean |
| 09 | Alto Paraná | 24°41′04″ S | 54°52′11″ W | Soybean |
| 10 | Alto Paraná | 25°00′01″ S | 54°52′59″ W | Soybean |
| 11 | Alto Paraná | 25°03′13″ S | 54°55′00″ W | Soybean |
| 12 | Alto Paraná | 25°08′01″ S | 54°58′02″ W | Soybean |
| 13 | Alto Paraná | 25°10′38″ S | 54°56′42″ W | Soybean |
| 14 | Alto Paraná | 25°37′56″ S | 54°58′16″ W | Soybean |
| 15 | Alto Paraná | 25°36′42″ S | 54°58′55″ W | Soybean |
| 16 | Alto Paraná | 25°54′16″ S | 55°07′03″ W | Soybean |
| 17 | Alto Paraná | 25°00′14″ S | 54°56′50″ W | Soybean |
| 18 | Canindeyú | 24°04′26″ S | 54°27′05″ W | Soybean |
| 19 | Canindeyú | 24°06′33″ S | 54°31′46″ W | Soybean |
| 20 | Canindeyú | 24°07′01″ S | 54°34′59″ W | Soybean |
| 21 | Canindeyú | 24°33′15″ S | 54°46′47″ W | Soybean |
| 22 | Alto Paraná | 25°02′23″ S | 54°54′18″ W | Soybean |
| 23 | Canindeyú | 24°21′03″ S | 55°02′29″ W | Soybean |
| 24 | Canindeyú | 24°04′08″ S | 54°49′33″ W | Soybean |
| 25 | Canindeyú | 24°15′57″ S | 54°43′34″ W | Soybean |
| 26 | Canindeyú | 24°20′19″ S | 55°00′56″ W | Soybean |
| 27 | Canindeyú | 24°03′34″ S | 55°00′20″ W | Soybean |
| 28 | Canindeyú | 24°09′26″ S | 54°52′21″ W | Oat |
| 29 | Canindeyú | 24°10′39″ S | 54°53′20″ W | Oat |
| 30 | Canindeyú | 24°12′00″ S | 54°56′02″ W | Oat |
| 31 | Alto Paraná | 25°45′04″ S | 55°04′39″ W | Oat |
| 32 | Canindeyú | 24°11′60″ S | 54°56′10″ W | Chia |
| 33 | Canindeyú | 24°08′58″ S | 54°51′24″ W | Pasture |

*2.2. Screening*

In a greenhouse, with daily irrigation, in the municipality of Katueté, Canindeyú, Paraguay (24°09′28.7″ S, 54°52′10.4″ W), about 100 seeds were sown in a plastic tray filled with substrate potting mix, for each sampling site, from October to November 2018. After germination, seedlings were transplanted into 800 mL plastic pots and filled with substrate potting mix with one seedling per pot. A completely randomized design with eight replications was used for each herbicide applied. Six herbicides, at the average recommended dose for the control of *C. sumatrensis* at the stage of 6–8 true leaves, were applied to plants, in addition to the control (no application) (Table 2), for each sampling site. The application took place at the stage of 6–8 true leaves, using a series 110.02 (TeeJet Technologies, Wheaton, IL, USA) $CO_2$ backpack sprayer pressurized at a constant pressure of 2 kgf cm$^{-2}$, with a bar with four fan nozzles, positioned at 50 cm from the target and at a speed of 1 m s$^{-1}$, providing a total spray volume of 200 L ha$^{-1}$.

Plant control was evaluated at 28 days after application (DAA), and visual scores were assigned to each experimental unit, where 0 represents no damage and 100% indicates total plant death [16]. The results were presented descriptively. After screening, plants from certain populations were selected to be grown alone to generate $F_1$ seeds, which were used for the construction of dose–response curves. The generation of $F_1$ is important to attest to the inheritance of the resistance character of populations.

**Table 2.** Herbicides applied to *C. sumatrensis* plants for each site.

| Herbicide | Group | Dose [1] | Commercial Product [2] |
|---|---|---|---|
| 2,4-D | O—synthetic auxins | 1005 | DMA® 6 |
| paraquat | D—photosystem I inhibitors | 400 | Tecnoquat® SL |
| glyphosate | G—EPSPs inhibitors | 720 | Roundup Full® II |
| chlorimuron | B—ALS inhibitors | 20 | Poker® 75 WG |
| saflufenacil | E—PPO inhibitors | 35 | Heat® |
| glufosinate | H—GS inhibitors | 500 | Finale® |
| control (without application) | - | - | - |

[1] Doses in g ae ha$^{-1}$, for glyphosate and 2,4-D. For the others, in g ai ha$^{-1}$. Recommended average dose for the control of *C. sumatrensis* at the stage of 6–8 true leaves. [2] DMA® 6, Dow AgroSciences Paraguay S.A., Asunción, Paraguay; Tecnoquat® SL, Tecnomyl S.A., Asunción, Paraguay; Roundup Full® II, Monsanto Paraguay S.A., Asunción, Paraguay; Poker® 75 WG, Glymax Paraguay S.A., Hernandarias, Paraguay; Heat®, BASF Paraguay S.A., Asunción, Paraguay; Finale®, BASF Paraguay S.A. Asunción, Paraguay

*2.3. Dose–Response Curves*

The same screening procedures were followed for sowing, seedling transplantation, and herbicide application, at the same location. The biotype whose $F_1$ seeds were collected and investigated for resistance came from sampling site 27 (24°03′34″S 55°00′20″W), and the susceptible biotype, also from the $F_1$ generation, came from site 33 (24°08′58″S 54°51′24″W). The herbicides applied were paraquat (0, 50, 100, 200, 400, 800, 1600, and 3200 g active ingredient (ai) ha$^{-1}$ (Tecnoquat® SL, Tecnomyl S.A., Asunción, Paraguay) combined with 0.1% (*v/v*) non-ionic adhesive spreader; glyphosate (0; 90; 180; 360; 720; 1440; 2880 and 5760 g acid equivalent (ae) ha$^{-1}$ (Roundup Full® II, Monsanto Paraguay S.A., Asunción, Paraguay); and chlorimuron (0, 2.5, 5, 10, 20, 40, 80, and 160 g ai ha$^{-1}$ (Poker® 75 WG, Glymax Paraguay S.A., Hernandarias, Paraguay) combined with 0.5% (*v/v*) mineral oil. The doses used represent the dose recommended in the package insert for each herbicide, in proportions of 0, $^1/_8$, $^1/_4$, $^1/_2$, 1, 2, 4, and 8X the recommendation. A completely randomized design was used with four replications, for each herbicide dose. Each repetition consisted of a 0.8 L plastic pot, with one plant per pot.

The application took place at the stage of 6–8 true leaves, via a backpack sprayer pressurized with $CO_2$, with a constant pressure of 2 kgf cm$^{-2}$, with a bar with four fan nozzles, series 110.02 (TeeJet Technologies, Wheaton, IL, USA) positioned at 50 cm from the target, and at a speed of 1 m s$^{-1}$, providing a total spray volume of 200 L ha$^{-1}$.

The plant control was evaluated at 28 DAA; visual scores were assigned to each experimental unit, where 0 indicates no damage and 100% indicates total plant death [16]. Dry mass evaluation was carried out at 28 DAA of the herbicides. Plants were cut at the ground level, placed in paper bags, dried in an oven at 70 °C for four days (to reach constant mass), and then measured.

*2.4. Statistical Analysis*

After screening for the generation of heritability ($F_1$), selection of biotypes, and realization of dose–response curves, the data of the evaluations from 28 DAA were subjected to analysis of variance and regression ($p \leq 0.05$) and adjusted for the nonlinear logistic regression model proposed by Streibig [17]:

$$y = a/[1 + (x/b)\hat{}c], \tag{1}$$

where y is the response variable (percentage control or shoot dry mass); x is the herbicide dose (g ha$^{-1}$); and a, b, and c are the estimated parameters of the equation, so that a is the amplitude between the maximum and the minimum point of the variable, b is the dose that provides 50% response, and c is the slope of the curve around b.

The non-linear logistic model provides an estimate of parameter $C_{50}$ (50% control) or $GR_{50}$ (50% growth reduction). Thus, we opted for mathematical calculation using the inverse equation of

Streibig [17], allowing the calculation of $C_{50}$, as proposed by Souza et al. [18]. The models used to obtain $C_{50}$ were the same as those used by Takano et al. [19], Takano et al. [20], and Albrecht et al. [14].

$$x = b(|a/y - 1|)\char`\^(1/c). \tag{2}$$

Based on the values of $C_{50}$ and $GR_{50}$, we calculated the resistance factor (RF = $C_{50}$ or $GR_{50}$ of the resistant biotype/$C_{50}$ or $GR_{50}$ of the susceptible biotype). The resistance factor expresses the number of times that the dose required to control 50% resistant biotypes is greater than the dose controlling 50% susceptible biotypes [15,21].

## 3. Results

For all sampling sites, 100% control of *C. sumatrensis* plants was found with the application of saflufenacil and glufosinate; for 2,4-D, the results were close to 100%. For paraquat, ≤50% control was observed in 19 out of the 33 sampling sites; for glyphosate, in 12 sites; for chlorimuron, in 7. The population of four sampling sites (13, 18, 25, and 27) had control ≤50% for the application of glyphosate, chlorimuron, and paraquat, simultaneously. In 12 sites, a control ≥86% was verified for paraquat, with only 2 sites for glyphosate, and only 2 for chlorimuron (Table 3).

**Table 3.** Control of the *C. sumatrensis* populations at 28 days after herbicide application.

| Site | Paraquat | Glyphosate | Chlorimuron | 2,4-D | Glufosinate | Saflufenacil | No Application |
|------|----------|------------|-------------|-------|-------------|--------------|----------------|
| | | | | ---%--- | | | |
| 1 | 100 | 65 | 70 | 100 | 100 | 100 | 0 |
| 2 | 90 | 70 | 80 | 100 | 100 | 100 | 0 |
| 3 | 70 | 65 | 65 | 98 | 100 | 100 | 0 |
| 4 | 45 | 70 | 65 | 100 | 100 | 100 | 0 |
| 5 | 100 | 60 | 75 | 100 | 100 | 100 | 0 |
| 6 | 95 | 75 | 60 | 100 | 100 | 100 | 0 |
| 7 | 25 | 65 | 65 | 100 | 100 | 100 | 0 |
| 8 | 30 | 60 | 65 | 100 | 100 | 100 | 0 |
| 9 | 60 | 45 | 45 | 95 | 100 | 100 | 0 |
| 10 | 15 | 40 | 60 | 100 | 100 | 100 | 0 |
| 11 | 95 | 50 | 40 | 100 | 100 | 100 | 0 |
| 12 | 20 | 55 | 55 | 95 | 100 | 100 | 0 |
| 13 | 20 | 40 | 50 | 100 | 100 | 100 | 0 |
| 14 | 15 | 45 | 70 | 100 | 100 | 100 | 0 |
| 15 | 15 | 60 | 65 | 100 | 100 | 100 | 0 |
| 16 | 95 | 60 | 60 | 100 | 100 | 100 | 0 |
| 17 | 100 | 65 | 60 | 100 | 100 | 100 | 0 |
| 18 | 30 | 35 | 45 | 100 | 100 | 100 | 0 |
| 19 | 95 | 65 | 40 | 98 | 100 | 100 | 0 |
| 20 | 100 | 50 | 70 | 100 | 100 | 100 | 0 |
| 21 | 20 | 55 | 55 | 100 | 100 | 100 | 0 |
| 22 | 20 | 55 | 55 | 95 | 100 | 100 | 0 |
| 23 | 40 | 50 | 60 | 98 | 100 | 100 | 0 |
| 24 | 50 | 60 | 60 | 100 | 100 | 100 | 0 |
| 25 | 20 | 50 | 50 | 100 | 100 | 100 | 0 |
| 26 | 100 | 100 | 100 | 100 | 100 | 100 | 0 |
| 27 | 15 | 30 | 40 | 95 | 100 | 100 | 0 |
| 28 | 25 | 60 | 60 | 100 | 100 | 100 | 0 |
| 29 | 40 | 40 | 65 | 100 | 100 | 100 | 0 |
| 30 | 30 | 55 | 65 | 95 | 100 | 100 | 0 |
| 31 | 25 | 45 | 55 | 98 | 100 | 100 | 0 |
| 32 | 100 | 50 | 70 | 100 | 100 | 100 | 0 |
| 33 | 100 | 100 | 100 | 100 | 100 | 100 | 0 |

Site 27 (resistant to paraquat, glyphosate, and chlorimuron) and site 33 (susceptible) used for dose–response curves.

Points with ≤50% control were plotted in red, from 51% to 85% in yellow, ≥86% in green, highlighting site 27 (R)—resistant to paraquat, glyphosate, and chlorimuron—and site 33 (S)—susceptible to herbicides. The proximity of collection points of the resistant and susceptible biotypes is presented in Figure 1.

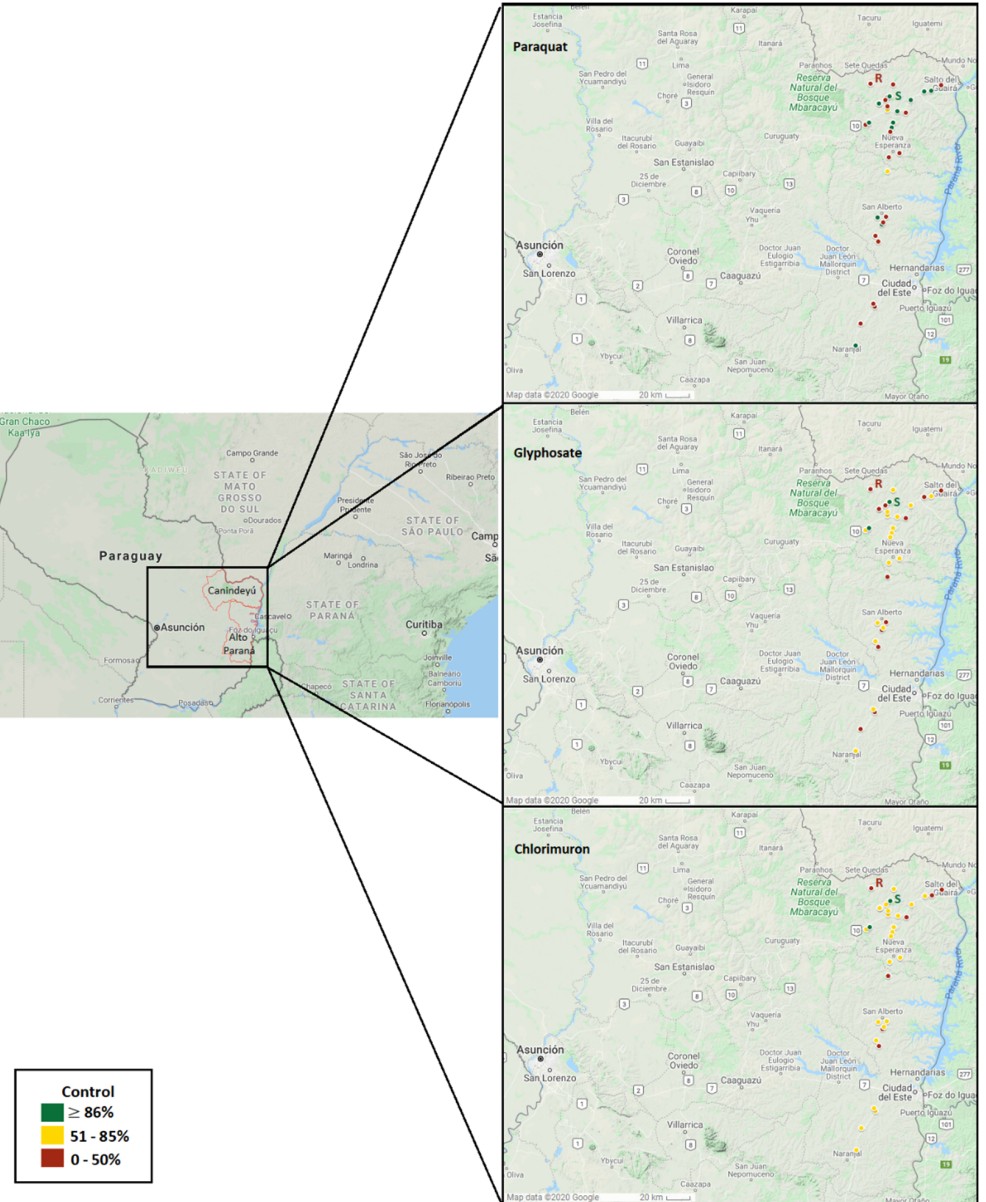

**Figure 1.** The geographic distribution of *C. sumatrensis* collection points with suspected resistance to herbicides, with respective effectiveness in the control. Canindeyú and Alto Paraná, Paraguay, 2017/18, season. R: site 27—resistant to paraquat, glyphosate, and chlorimuron; S: site 33 —susceptible.

According to the results of the screening, the biotype from site 27 was selected to investigate the possible case of resistance to herbicides. An RF of 6.79 was observed for $C_{50}$ (Figure 2A) and 3.92 for $GR_{50}$ (Figure 2B), for the application of paraquat. The ineffectiveness in controlling *C. sumatrensis* under the application of glyphosate was also verified; for $C_{50}$ and $GR_{50}$, RF was 12.32 (Figure 2C) and 4.15 (Figure 2D), respectively. For chlorimuron, RF was 11.32 for $C_{50}$ (Figure 2E) and 10.96 for $GR_{50}$ (Figure 2F). This confirmed the triple resistance of the *C. sumatrensis* biotype (site 27) to the herbicides paraquat, glyphosate, and chlorimuron (Table 4).

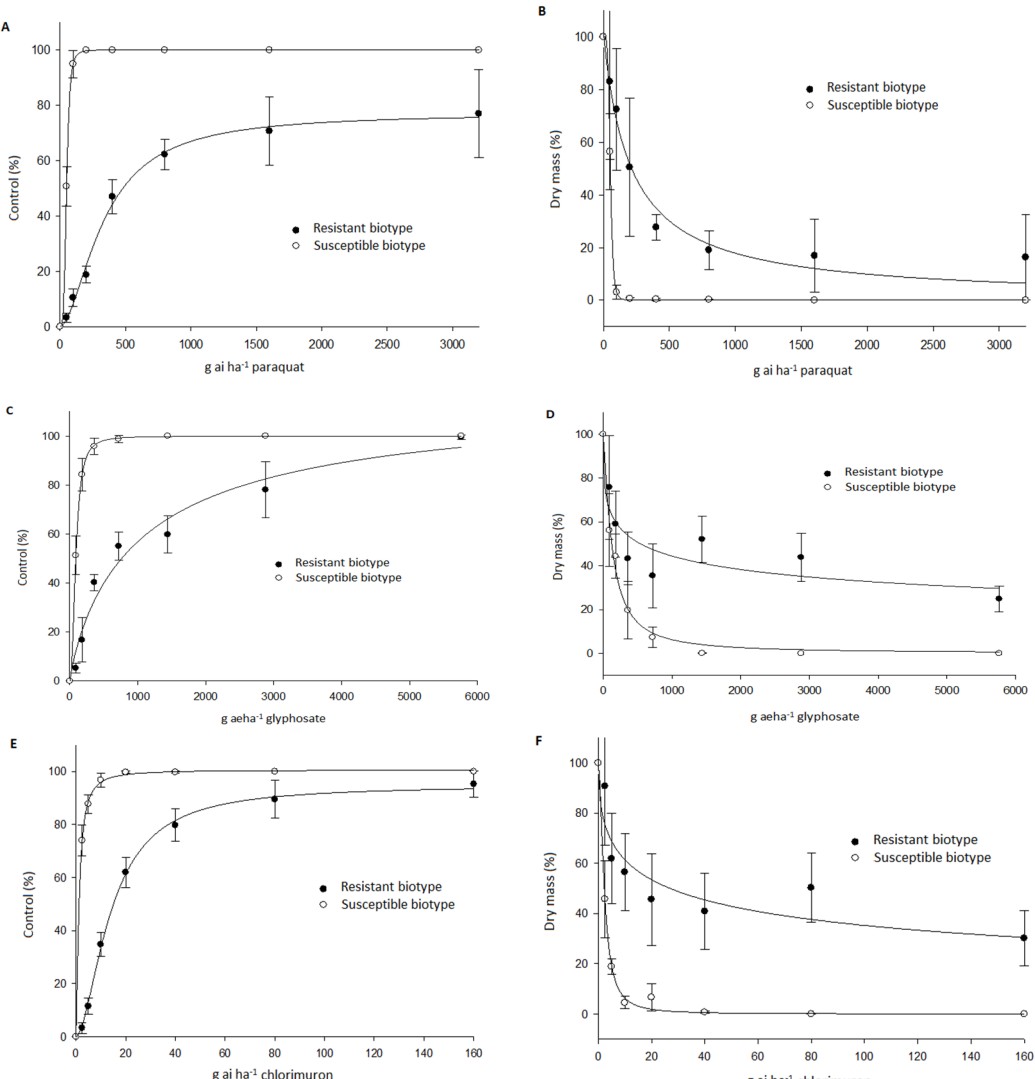

**Figure 2.** Control (**A**) and dry mass (**B**) of *C. sumatrensis* at 28 days after paraquat application. Control (**C**) and dry mass (**D**) of *C. sumatrensis* at 28 days after glyphosate application. Control (**E**) and dry mass (**F**) of *C. sumatrensis* (%) at 28 days after chlorimuron application. Site 27 (resistant to paraquat, glyphosate, and chlorimuron) and site 33 (susceptible). Bars shows the standard deviation (SD), *n* = 4.

**Table 4.** The required dose of herbicides for $C_{50}$ (50% control) or $GR_{50}$ (50% growth reduction) and resistance factor (RF) for *C. sumatrensis*.

| | Paraquat | | Glyphosate | | Chlorimuron | |
|---|---|---|---|---|---|---|
| **Biotype** | $C_{50}$ | $GR_{50}$ | $C_{50}$ | $GR_{50}$ | $C_{50}$ | $GR_{50}$ |
| | | | g ha$^{-1}$ | | | |
| Susceptible (site 33) | 49.65 | 52.46 | 87.85 | 126.10 | 1.25 | 2.26 |
| Resistant (site 27) | 337.19 | 205.94 | 1082.36 | 523.35 | 14.16 | 24.78 |
| RF | 6.79 | 3.92 | 12.32 | 4.15 | 11.32 | 10.96 |

Dose in g ai ha$^{-1}$ for paraquat and chlorimuron, in g ae ha$^{-1}$ for glyphosate.

## 4. Discussion

The low efficiency of paraquat, glyphosate, and chlorimuron was observed in most areas where *C. sumatrensis* seeds were collected. Control of ≥86% was observed in only two sites, for the three herbicides simultaneously. The identification of biotypes resistant to the three herbicides demonstrated the low effectiveness of these herbicides in controlling *C. sumatrensis* in a large area. The low effectiveness

of these herbicides against *C. sumatrensis* was been reported in Brazil, including in states bordering Paraguay (Paraná and Mato Grosso do Sul). This low efficacy was confirmed by the cases of simple and multiple resistance to paraquat, glyphosate, and chlorimuron [14,22,23]. Albrecht et al. [14] showed multiple resistance to paraquat, glyphosate, and chlorimuron with RF for the $C_{50}$ of 7.43, 3.58, and 14.35 and for the $GR_{50}$ of 2.65, 2.79, and 11.31, respectively. In the present study, we observed RF for the $C_{50}$ of 6.79, 12.32, and 11.32 and for the $GR_{50}$ of 3.92, 4.15, and 10.96, respectively, for paraquat, glyphosate, and chlorimuron—that is, with RF close to paraquat and chlorimuron in the comparison between these biotypes. A higher RF was found for glyphosate in the biotype identified in Paraguay in this study.

In contrast, the herbicides saflufenacil and glufosinate were effective in controlling *C. sumatrensis* in all sampling sites, and the herbicide 2,4-D also showed good control; however, 2,4-D and other synthetic auxins are the subject of other specific studies due to the rapid necrosis, as studied in Brazil [24]. This reinforces the need to use different herbicides to control weeds, focusing not only on management, but also on preventing the selection of new resistant biotypes. Other studies demonstrated the effectiveness of these herbicides in the control of species of the genus *Conyza* [25–29]—in most situations, in combination with other herbicides, including products with confirmed resistance.

The combination and rotation of herbicides with different mechanisms of action are reinforced by several studies as essential in preventing the selection of new cases, in the effective management of already resistant cases, and in expanding the spectrum of action of the herbicidal treatment [30–32]. In addition, non-chemical measures, such as cover crops, should be highlighted. For example, vetch and barley crop residues were effective in suppressing *C. canadensis* [33], and black oat and wheat in suppressing *C. bonariensis* [34]. The importance of monitoring the populations of resistant weeds is therefore emphasized, which allows for the identification of the evolution and dispersion of cases of resistance, which consequently provides subsidies for decision-making for the effective management of weeds [35,36]. This study highlights the importance of and identifies the levels of effectiveness of herbicides in the region where the biotype was recorded.

The monitoring weed resistance cases is, therefore, an essential practice to understand, identify, and quantify the frequency of these plants in advance [37]. Thus, studies on resistance monitoring lead to increased research and, consequently, new techniques for the control of problematic plants, such as the use of pre-emergent herbicides to decrease the selection pressure [38–40].

In Paraguay, only four cases of herbicide-resistant weed biotypes have been officially reported, including the present study. In addition to this, *Euphorbia heterophylla* was found to be resistant to imazethapyr (an ALS inhibitor), and *Digitaria insularis* and *Bidens subalternans* were resistant to glyphosate [1]. This reinforces the importance of the present study, not only by identifying the first case of multiple resistance in the country, but also for monitoring the population of *C. sumatrensis* and investigating the effectiveness of herbicides. This provides important information for the management of this weed and for prevention of the selection of new resistant biotypes.

This population of *C. sumatrensis* meets all the criteria set to confirm a new case of resistance to paraquat, glyphosate, and chlorimuron, according to the criteria for confirming a new case of weed resistance to a herbicide of the Herbicide Resistance Action Committee (HRAC) [41]. These criteria include the definition of weed resistance; confirmation of the results obtained by scientifically based protocols; characterization of the heritability of weed resistance to the herbicide; demonstration of the practical impact in the field of weed resistance to the herbicide; and botanical identification of the weed species under analysis and not as a result of deliberate/artificial selection. This case was reported to the International Herbicide-Resistant Weed Database and is already registered [1].

## 5. Conclusions

Our results confirmed the multiple drug resistance of *C. sumatrensis* to the herbicides paraquat (a photosystem I inhibitor), glyphosate (an EPSPs inhibitor), and chlorimuron (an ALS inhibitor) as all the criteria set to prove new cases of resistance of weeds were met, thus scientifically demonstrating the first case of a weed with multiple resistance to herbicides in Paraguay.

Population monitoring indicated the presence of *C. sumatrensis* plants with triple multiple resistance in the departments of Canindeyú and Alto Paraná, Paraguay, in most of the sampled sites. Further monitoring research on this weed species is ongoing in Paraguay, also covering the suspected resistance to 2,4-D and for other weed species, due to the scarcity of results in this country. Studies are underway with the objective of characterizing effective and sustainable alternatives for the control of this weed.

**Author Contributions:** Conceptualization and investigation, A.J.P.A., L.P.A., A.P., G.T., and F.S.A.; formal analysis, A.J.P.A., G.T., J.B.L., and M.T.Y.D.; data collection, A.J.P.A., G.T., and A.P.; writing—original draft, A.F.M.S.; writing—review and editing, all authors. All authors have read and agreed to the published version of the manuscript.

**Funding:** This research received no external funding.

**Conflicts of Interest:** The authors declare no conflict of interest.

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
