# Peer review of "Conyza sumatrensis Resistant to Paraquat, Glyphosate and Chlorimuron: Confirmation and Monitoring the First Case of Multiple Resistance in Paraguay"

_agriculture, doi:10.3390/agriculture10120582_

Round 1
Reviewer 1 Report
The manuscript has potential, however, needs to address certain concerns before considering acceptance. Specific comments/suggestions are annotated into comment boxes in the attached pdf file.
The scientific names should be italic within the abstract section. At the moment they are in normal font.

Reviewer 2 Report
Thank you for a very well designed research. It's was a pleasure to read the article.
I have noted some typo errors for eg:
line 21 'resistance for this specie" s missing in the species.
line 100--same local? is that meant to be same location?
F1 , 1 should be subscript not a normal 1
Line 106: was there supposed to be a comma (,), after Glymax?
I recommend a thorough spell check and typo check before the final submission
The second point I am interested in is, has there been research on the population genetic diversity of the study species in the sample collection locations? If there is, then it is worth mentioning in the introduction. If 'yes', were there any biotypes/ ecotypes detected within the populations? Do they have any correlation with the noted resistance?
Reviewer 3 Report
This paper identifies the first case of multiple herbicide resistance to paraquat, chlorimuron and glyphosate in biotypes of Conyza sumatrensis in Paraguay. Confirming the resistance of weeds to herbicides is always important to help inform management practices of the weed. Therefore this paper is significant for growers in Paraguay dealing with this weed.
I have two main issues with the paper in its current form. The first issue is the English. Much of the manuscript requires editing. The second issue is with the number of plants screened. You mention in section 2.2 that 100 seeds were sown in a pot and then transplanted to pots with a single seedling per pot. Later you mention there were eight replicates. It is not clear if this means 8 * 100 pots with a single plant or 8 pots each with a single plant. This should be better explained. If it is the latter then the experiment should be repeated with more replications. Additionally it is mentioned that there were four replicates in the dose-response experiment. It should be mentioned how many plants were included per replicate.
Additional comments for the methods and results sections:
- What was your criteria for selecting plants from the initial screening to be grown to produce the F1 generation? Was this due to low seed numbers collected?
- Your methods for the dose response experiment should be given their own section heading.
- Figure 1 is difficult to understand without context. The description should be rewritten to better explain that the figure shows the effectiveness of the herbicide in each image for the biotypes found at each location.
- For line 132 it would be better to say that paraquat controlled ≤ 50% of plants in those populations, rather than “a maximum of 50% control”.
- The descriptions of figures 1 and 2 should mention that the R population is from site 27 and the S population is from site 33.
- You need to discuss the effect that herbicide had on reducing biomass. As you calculated the GR50 for control you should also calculated this for biomass. Then include the resistance factor for biomass as well. It is important to know how much each herbicide reduces the biomass of the resistant biotype when it is unable to completely control it.
A few additional comments:
Line 21: typo for “species” (specie)
Line 58: Conyza should be written in full as it is the start of the sentence.
Line 88: Need to write in full the meaning of DAA here rather than on line 110.
Reviewer 4 Report
The present study investigated incidence of Conyza sumatrensis biotypes resistant to the herbicides glyphosate, chlorimuron and paraquat across Canindeyu and Alto Parana, Paraguay. Seeds were collected from field populations following failed herbicide application in soybean, oat, chia and pasture fields. Discriminating doses of paraquat, glyphosate, chlorimuron, 2,4-D, glufosinate and saflufenacil were used to screen for resistant and susceptible populations. Glufosinate, 2,4-D and saflufenacil effectively controlled all C. sumatrensis populations. Paraquat, glyphosate and chlorimuron failed to control (<50%) 19, 12 and 7 populations, respectively. One population (site 27) was selected for additional screening with paraquat, glyphosate and chlorimuron. This population had a GR50 of 3.92, 4.15 and 10.96 for paraquat, glyphosate and chlorimuron, respectively confirming resistance to these herbicides. The results are clearly presented and conclusions supported by the results, however, the materials & methods and discussion has several issues that need to be addressed.
What is being treated as an experimental unit? Is a single seedling per pot per population an experimental unit?
What type of nozzle is being used?
Any Teejet 11002 should be positioned 50 cm, not 500 cm above the target.
Need to describe how F1s were generated and with what populations.
The discussion needs work. The significance of this study is barely mentioned. Attention should be paid to (1) documented spread of poor control across this region, (2) documented resistance to 3 modes of action, and (3) the level of resistance documented in this study compared to the literature.
See specific comments in marked up manuscript
Round 2
Reviewer 3 Report
The manuscript has improved upon the points outlined in my previous report. I think that, if possible, the dose-response experiment should be repeated as four reps of one plant is a low number to analyze for a single herbicide dose. I think a second run of the dose-response experiment would strengthen the results.
Author Response
Response to Reviewer 3 Comments
Point 1: The manuscript has improved upon the points outlined in my previous report. I think that, if possible, the dose-response experiment should be repeated as four reps of one plant is a low number to analyze for a single herbicide dose. I think a second run of the dose-response experiment would strengthen the results.
Response 1: Dear editor, we appreciate your comment. But it is not possible to repeat the dose-response curves, which were carried out in 2018. We reiterate that the study follows international standards, the case is registered on the website The International Herbicide-Resistant Weed Database (weedscience.org). It should be noted that this population of C. sumatrensis meets all the criteria set to confirm a new case of resistance to paraquat, glyphosate and chlorimuron, according to the criteria for confirming the new case of weed resistance to an herbicide of HRAC. These criteria are: definition of weed resistance; confirmation by results obtained by scientifically based protocols; characterizing the heritability of weed resistance to the herbicide; demonstration of the practical impact in the field of weed resistance to the herbicide; botanical identification of the weed species under analysis, and not as a result of deliberate/artificial selection.
Our study followed these steps:
Observation of populations in the field;
Screening with 33 populations (33 collection sites x 7 treatments x 8 repetitions = 1848 pots);
F1 generation (from sites 27 and 33), biotypes used for the curves;
Finally, the dose-response curve (2 biotypes x 3 herbicides x 8 doses x 4 repetitions = 192 pots).
Our study followed the methodology of recent articles with this theme:
Albrecht, A.J.P.; Pereira, V.G.C.; Souza, C.N.Z.; Zobiole, L.H.S.; Albrecht, L.P.; Adegas, F.S. Multiple resistance of Conyza sumatrensis to three mechanisms of action of herbicides. Acta Sci. Agron. 2020, 42, e42485. [https://doi.org/10.4025/actasciagron.v42i1.42485]
Takano, H.K.; Oliveira Júnior, R.S.; Constantin, J.; Braz, G.B.P.; Franchini, L.H.M.; Burgos, N.R. Multiple resistance to atrazine and imazethapyr in hairy beggarticks (Bidens pilosa). Cienc. Agrotecnol. 2016, 40, 548-554. [http://dx.doi.org/10.1590/1413-70542016405022316]
Takano, H.K.; Oliveira Júnior, R.S.; Constantin, J.; Braz, G.B.P.; Gheno, E.A. Goosegrass resistant to glyphosate in Brazil. Planta Daninha 2017, 35, e017163071. [http://dx.doi.org/10.1590/s0100-83582017350100013]
Regarding the language, the manuscript will be forwarded for review again. This time for the service offered by MDPI.